# Signalling adjustments to direct and indirect environmental effects on signal perception in meerkats

**Pauline Toni** 1☯, **Gabriella E. C. Gall** 1,2☯*, **Tim H. Clutton-Brock**1,3,4, **Marta B. Manser**1,2,4

**1** Kalahari Meerkat Project, Kuruman River Reserve, Northern Cape, South Africa, **2** Department of Evolutionary Biology and Environmental Studies, University of Zurich, Zurich, Switzerland, **3** LARG, Department of Zoology, Cambridge, United Kingdom, **4** Mammal Research Institute, University of Pretoria, Pretoria, South Africa

☯ These authors contributed equally to this work.
* gabriella.gall@yahoo.de

**Data Availability Statement:** Data has been reposited on dryad and can be accessed via doi:10. 5061/dryad.0p2ngf1ws.

## Abstract

The efficiency of communication between animals is determined by the perception range of signals. With changes in the environment, signal transmission between a sender and a receiver can be influenced both directly, where the signal's propagation quality itself is affected, and indirectly where the senders or receivers' behaviour is impaired, impacting for example the distance between them. Here we investigated how meerkats (*Suricata suricatta*) in the Kalahari Desert adjust to these challenges in the context of maintaining group cohesion through contact calls. We found that meerkats changed their calling rate when signal transmission was affected indirectly due to increased dispersion of group members as during a drought, but not under typical wet conditions, when signal transmission was directly affected due to higher vegetation density. Instead under these wetter conditions, meerkats remained within proximity to each other. Overall, both direct and indirect environmental effects on signal perception resulted in an increased probability of groups splitting. In conclusion, we provide evidence that social animals can flexibly adjust their vocal coordination behaviour to cope with direct and indirect effects of the environment on signal perception, but these adjustments have limitations.

## Introduction

Communication efficiency between animals is determined by the perception range of signals [1]. The perception range of acoustic signals depends on both the acoustic properties of the signal and the environment, specifically the physical properties and the acoustic background of the area between sender and receiver which the signal propagates through [1–3]. Changes in the environment can influence signal transmission both directly, where the signal itself is affected and indirectly, where a behaviour affecting signal production or perception is constrained. High vegetation density, for example, has a direct effect as it will increase sound attenuation and thus decrease the perception range [4–7]. The sound landscape, including

**Funding:** PT and GG were funded by the Swiss National Science Foundation (Grant No PDFMP3_141768 to MBM), MBM by the University of Zurich, and THCB was funded by the University of Cambridge. This paper has relied on records of individual identities and/or life histories maintained by the KMP, which has been supported financially by the European Research Council (Grant No 294494 to T.H. Clutton-Brock since 1/7/ 2012) and the University of Zurich, as well as logistically by the Mammal Research Institute of the University of Pretoria.

**Competing interests:** The authors have declared that no competing interests exist.

sounds produced by other animals, wind or anthropogenic noise, can have similar direct effects by masking the vocal signal [8–11]. In contrast a reduction in food distribution can pose an indirect effect, as it can modify the distance between sender and receiver [12,13], forcing them to forage at greater distance, potentially out of their normal hearing range. To mitigate the direct effects of the environment on the signal itself, senders can either reduce the distance to the receiver, adjust the amplitude and structure of the signal [2,6,9,14], or increase the production rate of the signal [2,6,15,16]. To diminish indirect environmental effects that influence inter-individual spacing, the sender cannot reduce spacing. Instead he can only try to adjust the signal production, including amplitude, signal structure and the production rate of a signal. The production rate of a signal matters especially for social animals on the move, e.g. olive baboons (*Papio anubis*) [6,17], goats (*Capra aegagrus hircus*) [18] or dolphins (*Tursiops truncatus*) [16], as they may frequently move in and out of the signals' perception range. Calling at higher rates therefore increases the probability of the signal being heard by the receiver at any given moment. The receiver can in both cases attempt to increase signal perception by reducing the distance to the sender, a mitigation option that has some limits in the case of indirect effects. When maintaining cohesion and remaining within given distances is a necessity, mediating the effects of changes in the environment will therefore mostly depend on the sender. For acoustic signals we accordingly expect changes in either signal amplitude, structure or signalling rate [2]. Here we are particularly interested in the effect of direct and indirect environmental changes on individual calling rate as well as on group cohesion.

Contact calls play a major role in group signalling in many mammal and bird species [19,20], and are either used to maintain group cohesion [21,22], or to space out group members to avoid competition [23]. Consequently, animals occupying environments with unpredictable rainfall, such as savannas or semi-deserts, and forming highly social and cohesive groups, e.g. baboons (*Papio*), vervet monkeys (*Chorocebus pygerythrus*), meerkats (*Suricata suricatta*), dwarf mongooses (*Helogale parvula*) or pied babblers (*Turdoides bicolor*) to name a few, should have adapted their contact signalling to a variety of environmental conditions, whereby the specific adaptations may differ depending of the specific function of the contact calls in a system. For such species, the direct environmental effects on signal perception will mostly be caused by changes in vegetation density. For example, during wet summer periods vegetation in dry savanna habitats is a lot denser and higher than during dry summer periods [24–28]. Background noise, such as wind [10,11] or the presence of noisy sympatric living species [29–31] and/or conspecifics [32], are further factors that can influence signal perception directly. Indirect effects on signal perception will mostly relate to the spacing among conspecifics. Seasonal changes in food distribution, and in particular droughts might lead to animals moving further away from their conspecifics in search for scarcer food. Importantly, both direct and indirect effects on call perception can lead to an increased risk of group fission and thus a decrease in foraging time as well as an increase in predation risk [33,34].

We investigated how individual group members maintain group cohesion and adjust their contact call rate to varying environmental conditions in cooperatively breeding meerkats. Meerkats are obligate group living mammals and inhabit areas of the dry, southern part of Africa with unpredictable rainfall, varying in intensity, duration and timing within and among years [35]. Groups of meerkats forage as a cohesive unit, constantly moving, mostly with the head directed downwards, in search for mainly small invertebrate prey in the sand [36]. As invertebrate abundance is greatly dependent on precipitation [36], meerkats are highly susceptible to food shortages due to droughts, here defined as hotter and drier periods lasting longer than the dry periods typically observed during this season. While foraging, close calls, low amplitude contact calls, are the most frequently emitted vocalization, functioning to maintain group cohesion [21,37]. When pups are present, meerkats reduce their close call rate,

presumably to avoid attracting the attention of begging pups or to avoid information redundancy, and might thus use the much louder pup vocalizations to localize the centre of the foraging group [32].

With meerkats relying on vocal exchanges to maintain group cohesion, and group cohesion being crucial for survival [38], we expect the animals to have adapted their vocal production to both direct and indirect environmental effects on signal perception. Specifically, we compared the effect of typical wet and dry conditions and an extreme drought on meerkat contact call rate and overall group cohesion. We predicted (i) an increase in distance to the nearest neighbour as a direct response to more scarcely distributed prey during the drought condition, but not during wet conditions, compared to dry conditions, and (ii) an increase in individual call rate under wet conditions to counteract direct effects on signal perception, and an even greater increase during the drought condition, to minimise separation risk compared to dry conditions. We used focal recordings collected during a wet, a dry and a drought year over the same months (October to February), to determine an individual's close call rate and the distance to its nearest neighbour during the different environmental conditions. With meerkats reducing their call rate substantially when pups are present likely to adjust to social noisiness, we controlled for the presence of pups during the different environmental conditions [32]. Since the distribution of food in the habitat could not be measured directly, we used changes in individual's morning body weight and weight gain–a proxy for morning foraging success–from long-term data collected by the Kalahari Meerkat Project as an indirect measure for changes in food abundance and thus to establish the presence of an indirect environmental effect on signalling behaviour. We (iii) predicted a linear decrease in body condition and foraging success from wet, to dry, to drought conditions due to increasingly scarcer food. We also analysed long-term data to explore whether (iv) group splits occurred more frequently during drought and wet conditions compared to dry conditions, as a potential result of changes in each individual's signal perception range. We expected more group splitting events during the drought condition than during wet conditions, as in contrast to wet conditions signal perception is reduced indirectly through an increase in spacing, and thus receivers are limited to contribute to improve signal perception by reducing the distance to the sender.

## Methods

The data for this study were collected at the Kalahari Meerkat Project (KMP), Kuruman River Reserve, in the Northern Cape Province, South Africa (26˚58' S, 21˚49' E). Descriptions of the general climate and habitat are provided by Clutton-Brock et al [35]. For this study, we were particularly interested in the summer period (October to February) and summarized the meteorological information for these periods from the last seven years in Table 1. Based on the amount of rainfall, the mean maximum and minimum daily temperature and the mean normalized difference vegetation index (NDVI) recorded for the study site (Table 1), we assigned each of the summer periods to one of three distinct environmental conditions: 'wet', 'dry' or 'drought'. This resulted in three summer periods with wet conditions (2010/2011, 2011/2012 and 2016/2017) described by high levels of rainfall, relatively low temperatures and relatively high NDVI score, three summer periods with dry conditions (2012/2013, 2013/2014 and 2014/2015), described by medium rainfall, high temperatures and a low NDVI score and one summer period with drought conditions (2015/2016), described by very low rainfall, very high temperatures and a low NDVI score (Table 1). As part of the long-term, longitudinal data collection at the KMP, each meerkat group was visited three to four days per week in the morning and/or the evening [39]. We analysed long-term data on body condition, foraging success and the occurrence of group splits for the summer periods between October and February 2011

**Table 1. Summary of the meteorological data and the Normalized Difference Vegetation Index (NDVI) for each period of data collection and the resulting categorisation into the environmental conditions used for the different analyses.**

| Period (Oct to Feb) | Environmental Condition | Number of rain days | Total rain (mm³/m) | Max daily rain (mm³/m) | Average max daily temperature (˚C) | Average min daily temperature (˚C) | Average NDVI score |
|---|---|---|---|---|---|---|---|
| 2010/2011 | Wet | 46 | 321.2 | 31.8 | 34.1 | 15.9 | 0.26 |
| 2011/2012 | Wet | 26 | 176.8 | 28.4 | 34.4 | 13.8 | 0.22 |
| 2012/2013 | Dry | 17 | 33.0 | 10.0 | 36.2 | 16.1 | 0.16 |
| 2013/2014 | Dry | 26 | 98.8 | 23.8 | 34.2 | 15.8 | 0.19 |
| 2014/2015 | Dry | 28 | 50.8 | 9.2 | 35.4 | 15.6 | 0.16 |
| 2015/2016 | Drought | 9 | 13.4 | 5.8 | 36.3 | 17.5 | 0.15 |
| 2016/2017 | Wet | 25 | 248.0 | 47.0 | 34.7 | 16.0 | 0.21 |

until 2017. Furthermore, we collected data for a direct comparison of vocal behaviour and spatial organisation for the summer periods between October and February from 2014 to 2017, with each of the years representing one of the three environmental conditions. All animals in the population were habituated to close human observation up to 1 m and could be identified through individual dye mark combinations [40]. As the number of individuals and groups vary between the different summer periods, we summarized the group compositions in Table 2 for each of the following analyses separately. For all analysis, we only included the visits to the group during the five months of each summer period specified above.

## Ethical note

All data collection adhered to the Association for the Study of Animal Behaviour (ASAB) guidelines. All research was conducted under the permission of the ethical committee of Pretoria University (Permit number: EC031-13) and the Northern Cape Conservation Service, (FAUNA 1020/2016), South Africa. Access to the field site was granted by the Kalahari Research Trust as well as the neighbouring farmers.

**Table 2. Description of groups used for each period, categorized in environmental conditions, and total number of groups and individuals used for each of the analyses.**

| Period (Oct to Feb) Environmental condition | | Total groups in long-term data | | | Number of groups/ individuals for each analysis | | | | | | | |
|---|---|---|---|---|---|---|---|---|---|---|---|---|
| | | Total number of groups | Mean ± SD group size | Range of group sizes | Morning weight and weight gain | | Speed | Group split | Vocal recordings of close calls and distances to nearest neighbour | | | |
| | | | | | Number of groups | Number of individuals | Number of groups | Number of groups | Number of groups | Number of Individuals | Number Groups with pups | Mean ± SD pups per group |
| 2010/2011 | Wet | 14 | 19 ± 6 | 3–34 | 14 | 207 | 14 | 14 | - | - | - | - |
| 2011/2012 | Wet | 16 | 19 ± 9 | 3–36 | 16 | 235 | 16 | 16 | - | - | - | - |
| 2012/2013 | Dry | 18 | 13 ± 7 | 3–36 | 18 | 264 | 18 | 18 | - | - | - | - |
| 2013/2014 | Dry | 17 | 9 ± 4 | 3–24 | 16 | 129 | 17 | 16 | - | - | - | - |
| 2014/2015 | Dry | 18 | 13 ± 5 | 3–24 | 17 | 109 | 18 | 18 | 12 | 71 | 9 | 3 ± 2 |
| 2015/2016 | Drought | 14 | 11 ± 5 | 3–24 | 14 | 119 | 14 | 14 | 11 | 63 | 6 | 3 ± 1 |
| 2016/2017 | Wet | 13 | 12 ± 6 | 3–25 | 12 | 31 | 13 | 13 | 10 | 31 | 3 | 4 ± 2 |

## Focal recordings of call rate and distance to nearest neighbour

To investigate whether the cohesion mechanism of meerkats was affected by environmental conditions, we collected detailed vocal and spatial information on the adult individuals of each meerkat group (see Table 2 for information on the number of individuals and groups) for one summer per condition. Vocalizations of the meerkats were recorded using a directional microphone (Sennheiser ME66 with K6 powering module) at 0.3–1.5 m to the focal individual, while simultaneously documenting the nearest neighbour identity and distances through a handheld microphone (Philips SBC MD 110) onto the second channel of a Marantz recorder (PMD661 professional, sampling frequency 48 kHz, 24 bit). Distances to the nearest neighbour were estimated visually and categorically ('0–2 m', '2–5 m', '5–10 m' and '>10 m') by the observer, and were documented whenever a change in category occurred. Due to the relatively small distances to be estimated, these categories provide fairly accurate data. The length of each recording was determined by the number of close calls emitted by the focal within the first 5 min of the recording: if less than 10 close calls were emitted, the recording was extended until 10 calls were recorded or to a maximum 10 min recording time. The number of close calls emitted by the focal and the time periods the focal spent at the different distance categories to its nearest neighbour were quantified from the sound files using Cool Edit Pro 2.0 (Syntrillium Software Corporation). We counted the overall number of close calls emitted by the focal during each recording (Table 2) and calculated the close call rate per minute foraging as well as the call rate per minute of the focal for each nearest neighbour distance category. Finally, we calculated the proportion of the total recording time each focal spent within each specific nearest neighbour distance categories.

## Frequency of group splits

To investigate whether reduced group cohesion may lead to higher frequency of group splits and varied depending on the environmental conditions, we analysed the long-term data of the KMP over a period of seven consecutive summers (Table 2). During each visit to a meerkat group, observers recorded if a group split had occurred. Group splits were defined as a group temporarily splitting into two or more mixed-sex subgroups being further than 100 m apart from each other for at least 15 min. Groups might travel at higher speeds and further distances during drought conditions compared with the wet or dry conditions, and this might additionally influence an individual's perception range. Accordingly, we also tested whether meerkats travelled further and at different speeds during under different conditions, but found no biologically meaningful difference (see S2 File).

## Body condition and foraging success

We used morning weight and foraging success as a proxy to establish whether drought conditions represent an indirect effect on signal perception. As part of the general protocol of the long-term data collection of the KMP each meerkat was weighed by climbing onto electronic scales where it was rewarded with a small amount of boiled egg or water. Meerkats of a particular group were weighed at each visit by a researcher to that group, before the group started foraging in the morning (morning weight), again at 'lunch' after about three hours of foraging (lunch weight), and in the evening before meerkats went below into their sleeping burrow (evening weight) [39]. To avoid any confounding effect due to the difference in dominant and subordinate individuals' weights, we only included adult subordinate individuals in our analysis and used the morning weights as a measure of body condition. Furthermore, we estimated an individual's foraging success by calculating its weight gain during the morning visit. To do so we calculated the difference between an individual's lunch weight and its morning weight

and standardized it by the time between the morning weight and the lunch weight for each individual. Positive values indicate an individual weight gain (g/hour), while values below zero reflect an individual weight loss.

## Statistical analyses

Statistical tests were carried out using R (version 3.5.0) [41]. To investigate whether the call rate changed depending on the environmental condition, the distance to the nearest neighbour, or the interaction between the two, we fitted a linear mixed-effects model (LMM) [42], also controlling for the effect of the presence of pups. We used the number of close calls emitted by a focal individual when in a specific distance category from its nearest neighbour as the response variable and added an offset of the logarithm of the total recording time. As explanatory variables, we included the interaction between the environmental condition and the distance category. For this analysis we combined the nearest neighbour distance categories '0–2 m' and '2–5 m' into '0–5 m; and '5–10 m' and '>10 m' into '>5 m' to reduce the number of factor levels and thus be able to fit the interaction in the GLMM. We included the identity of the focal individual nested within recording date and group identity in the random terms, as most of the individuals were recorded multiple times and most group members were recorded on the same date. To assess the significance of differences between the dry and the drought condition for this and each of the following analysis, we performed pairwise post-hoc comparisons using the package multcomp [43].

We examined whether meerkats spent a higher proportion of time further away from their nearest neighbour during the drought compared to the wet and dry conditions using a Dirichlet regression [44]. We used a matrix of the different time proportions within each nearest neighbour distance category as a response, and environmental condition as explanatory variable. We analysed the propensity for groups to split into subgroups depending on the environmental condition. For this we used a GLMM with a binomial link function, with the presence or absence of a split on a given observation day as the response variable, environmental condition as well as total group size as explanatory variables and group identity as random term.

To evaluate whether drought conditions influenced meerkat signalling indirectly, we investigated whether the body condition of meerkats was negatively influenced by the drought compared to wet and dry conditions, by fitting a LMM with an individual's morning weight as response variable and environmental condition as well as each individuals age as explanatory variable. As random factors we added summer period, to correct for the difference in number of summers comprised in each environmental condition (see Table 2 for details) as well as individual identity nested within group identity, to account for multiple measurements of individuals within each group. We did a stepwise model reduction, where the full model was compared to all lower level models. The models were compared based on Akaike information criterion (AIC) and we used the model with the lowest value as the model with the best fit. If the difference between models was within 2 delta AIC, we chose the one with the lower number of degrees of freedom as the best model. To assess the significance of each of the fixed effects in the final model we did a log likelihood ratio test (LRT), comparing for each fixed effect the final model to a model without the variable. The same process was also used for each of the following analyses that used linear or generalized (GLMM) linear mixed-effects models. In addition, for each model we checked explanatory variables' collinearity and visually inspected whether the model assumptions were fulfilled. To investigate whether foraging success of the meerkats was lower during the drought compared to the wet and dry conditions, we fitted a LMM with the standardized weight gain as response variable and environmental condition and individual age as explanatory variable and summer period, as well as individual

identity nested within group identity as random terms. We also tested whether individual age differed between environmental conditions by fitting a GLMM with the rounded age in years as response variable, environmental condition as explanatory variable and summer period as well as individual identity nested within group identity as random terms.

## Results

Increasing dryness affected foraging behaviour and the cohesion of meerkat groups substantially. In line with our predictions that meerkats might counter an increase in group dispersion under extreme environmental conditions with more frequent signal production, we found a significant interaction between the environmental condition and the distance to the nearest neighbour, whereby meerkats emitted more calls during the drought (6.50 ± 1.26 calls/min) than during wet (4.05 ± 1.34 calls/min, P = 0.653) and dry conditions (2.67 ± 1.24 calls/min, P = 0.003; Fig 1a) even at further distances to their nearest neighbour (LRT: $\chi^2$ = 11.7, P value = 0.008; S1 and S2 Tables in S1 File). Individuals emitted 4.36 ± 5.03 calls/min) when pups were present and 7.62 ± 9.00 calls/min when no pups were present (LRT: $\chi^2$ = 3.75, P value = 0.055; S1 Table in S1 File), suggesting a strong tendency for the presence of pups to decrease call rate, regardless of the environmental conditions.

When comparing the proportion of time meerkats spent within specific distance categories (0–2 m, 2–5 m, 5–10 m, >10 m) to their nearest neighbour, we found as expected that meerkats decreased the proportion of time spent within 0–2 m to their nearest neighbour from 68% in the wet to 62% in the dry and to 52% in the drought conditions. During the drought (32%), but not dry conditions (28%), the animals significantly increased the proportion of time they spent within 2–5 m to their nearest neighbour compared to the wet conditions (19%). In contrast to our predictions, however, there was no significant difference in the proportion of time individuals spent within 5–10 m when comparing the wet (9%) to the dry (8%) and the drought (13%) as well as the dry to the drought conditions. Similarly we found no difference in the proportion of time individuals spent further than 10m from their nearest neighbour when comparing between the wet (4%) and the dry (2%), the wet and the drought (3%) and the dry and the drought conditions (P = 0.80; S3 Table in S1 File, Fig 1b).

Groups were 2.2 times more likely to split in the wet conditions (2.69% splits within 2784 group visits) and 3.2 times more likely to split in the drought conditions (3.84% splits within 2225 group visits) compared to the dry conditions (1.21% splits within 4086 group visits, LRT: $\chi^2$ = 28.45, P value < 0.001, Fig 1c, S4 and S5 Tables in S1 File). In addition, the likelihood for groups to split increased with increasing group size (LRT: $\chi^2$ = 56.59, P value < 0.001, S4 and S5 Tables in S1 File).

Morning weight, taken as a proxy for body condition, was significantly lower during the drought condition (estimated morning weight = 520 ± 19.05 g) compared to wet conditions (average morning weight = 582 ± 20.82 g), but not compared to dry conditions (average morning weight = 560 ± 20.69 g). As morning weight during the dry periods was also not significantly lower than during the wet period this suggests a gradual decline of body condition with increasing dryness (LRT: $\chi^2$ = 8.13, P value = 0.02, S6 and S7 Tables in S1 File, Fig 1d). In contrast, we found no significant difference in foraging success, estimated through the standardized difference between an individual's lunch weight and its morning weight, between the wet (estimated weight gain = 6.05 ± 0.56 g/hour) and dry (estimated weight gain = 7.24 ± 0.75 g/hour), the wet and drought (estimated weight gain = 6.36 ± 1.06 g/hour) and the dry and drought conditions (LRT: $\chi^2$ = 3.22, P value = 0.20, Fig 1a, S8 and S9 Tables in S1 File). Both individual morning weight (LRT: $\chi^2$ = 652.7, P value < 0.001, S6 Table in S1 File) and foraging success (LRT: $\chi^2$ = 46.7, P value < 0.001, S8 Table in S1 File) were significantly correlated to

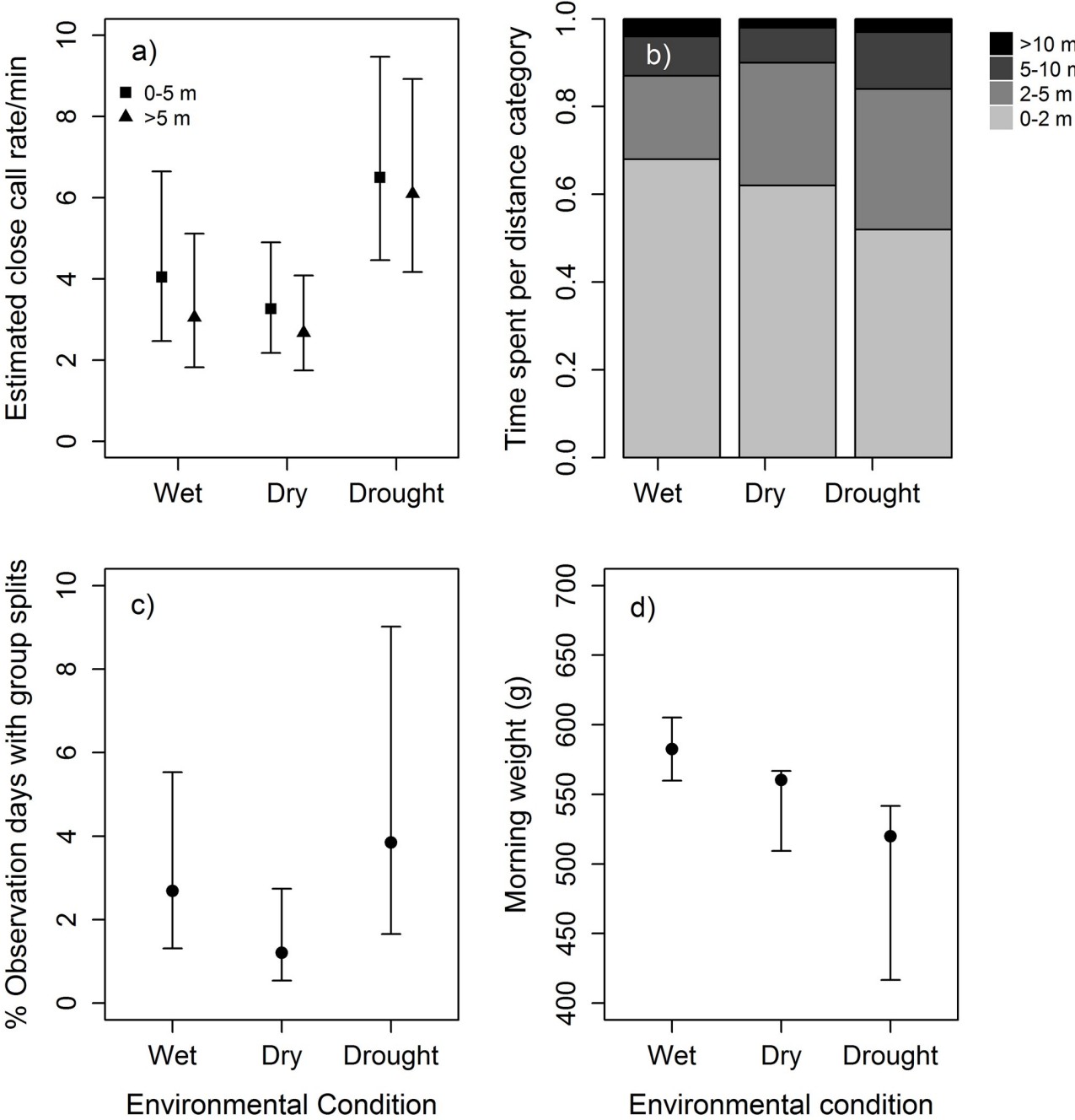

**Fig 1. (a) Effects of environmental condition on the close call rate of an individual when the nearest neighbour was within 5 m (circle) and further than 5 m (triangle) of the caller; (b) Proportion of time individuals spent at different distance categories ('0–2 m', '2–5 m', 5–10 m, and '>10 m') from their nearest neighbour during the different environmental conditions; (c) Percentage of observation days with group splits and (d) Mean morning weight of adult subordinate individuals under the different environmental conditions tested.** In panel a), c) and d) points represent (Generalised) Linear Mixed-effects Model estimates and error bars the 95% confidence intervals, while in panel b) the raw data is plotted.

the individuals age, but there was no difference in individual age between the wet (estimated age = 3.03 ± 1.33) and dry (estimated age = 3.81 ± 1.46), the wet and drought (estimated age = 6.84 ± 1.65) and the dry and drought conditions (LRT: $\chi^2$ = 1.65, P value = 0.44, S10 and S11 Tables in S1 File).

## Discussion

Our study shows that meerkats adjusted their vocal behaviour to changes in their environment and maintained high levels of group cohesion. Individual close call rate was significantly higher in the seldom occurring extreme drought periods compared to the more typically occurring wet and dry conditions. Meerkats spent less time within 0–2 m to their nearest neighbour in both the dry and the drought conditions than in the wet condition, and more time within 2–5 m in the drought, but not during the dry conditions. We found no effect of the environmental condition on the proportion of time spent at larger distances (5–10 m and >10 m) to the nearest neighbour. These findings are in line with our predictions that during drought conditions, when perception range is affected indirectly through increases in spacing among group members, the improvement of signal perception depends mostly on the sender. Accordingly, indirect effects of the environment will lead to changes in the property of the signal or the rate of its production as shown here. This is because group members likely try to avoid foraging competition from their conspecifics and search for more scattered and scarce food items, thus unavoidably increasing the distance between close neighbours. In contrast, when the perception range of a signal is affected directly, as here during the wet conditions with little food limitation, both the sender and the receiver can mediate the effect by reducing the distance to each other. Consequently, the properties of the signal or the production rate should be less affected. However, previous studies e.g. on baboons [6] and flycatchers [8] among many others, have shown that individuals adjust the production rate and/or modulate the structure of their vocalizations in response to increased vegetation density or noise [2,9]. This indicates that, in our study, the difference in vegetation density between the analysed wet and dry condition was likely not as influential as in these other studies and thus no additional changes in call production rate were necessary.

As we had no data available on the distribution and amount of prey items available under the different conditions, we used the changes in individual morning weight and weight gain as a proxy for changes in food abundance, to further support our assumption that environmental conditions could have indirect effects on signal perception. Morning weight declined gradually from the wet to the drought condition. We found no indication that the decline in morning weight was due to a difference in average individual age during the different conditions. Rather changes in morning weight may be explained by the effort to prevent rises in body temperature during extremely hot days, either with an increase in evaporative water loss or by moving into the shade and stopping to forage [45]. This result indicates a decrease in food abundance with increasing dryness and thus the presence of an indirect environmental effect on the perception range of signals. Individual weight gain per hour foraging remained similar regardless of environmental conditions. Meerkats experiencing environmental conditions with high food availability and temperatures not exceeding their body temperature, usually forage only until they are satiated and then spend their time on other activities, e.g. as sentinels [46] or resting and socialising (unpublished long-term data). In contrast during the drought condition, meerkats focused on more efficient foraging, thus maintaining a similar food intake per hour, and showed less cooperative sentinel behaviour in the drought compared to wet and dry conditions [47]. It is also important to note that we measured the weight gain (in g/hour) of individuals as proxy for foraging success and not the actual caloric intake, which may have differed between the different environmental conditions.

Group splits occurred most frequently during the drought and least frequently during dry conditions. This is in line with our predictions, that individuals are less able to maintain cohesion when signal perception is indirectly affected by the environment through an increase in individual spacing as is likely the case during the drought [24–26]. The cohesion mechanism

of meerkats functions by individuals adjusting their call rate depending on their own location within their group, and by following in the direction of 'vocal hotspots', areas where many calls are heard from [21]. During the dry conditions, meerkats were likely not food limited and following a global hotspot was possible. However, during the drought as individuals increase their call rate in order to improve signal perception several 'vocal hotspots' might emerge. As a result, while cohesion with the nearest neighbours was maintained, small subgroups foraging at increasing distances seemed to form (personal observations Toni & Manser). Following a single global hotspot may not be feasible once group members disperse over too large an area to remain within the hearing range of the signal of the majority of the group [21]. Instead, when group dispersion increases as in larger groups or also in smaller groups during a drought, each meerkat may only listen to its local environment rather than to the global call pattern, leading to the observed increased group fission. Foraging in smaller subgroups might be the optimal solution to cope both with an increase in predation risk and with the increased competition for resources between group members. With fission-fusion dynamics likely being a response to the cost of grouping [48–50], this result indicates that the optimal group size for communication decreases and that the cost of grouping increases when spacing becomes constrained and individuals are forced to forage at larger distances.

How a species will adjust its signalling behaviour depends on both the context and the function of a signal. While contact calls are one of the most frequent vocalizations of many birds and mammals, they vary in their function [19]. For example, meerkats are attracted to the contact calls and use vocal hotspots to follow in the direction where most calls are heard from. The sympatric living pied babblers, also foraging as cohesive groups on the ground, however, emit chucks (a type of contact call) at higher rates to reduce the likelihood of being approached by a conspecific, likely to avoid foraging competition [51]. Due to this different function of the contact calls of babblers–and this is purely speculative and would require to be tested to be confirmed–, the effect of environmental variation from wet to dry to drought conditions may be the opposite to what is seen in meerkats. During wet conditions with high food availability, babblers might therefore call at lower rates, while drought conditions might not affect chuck call rate at all, as animals are already much more spread out.

Our results provide evidence that social animals can flexibly adjust their group coordination behaviour to cope with direct and indirect effects of the environment on signal perception. Nevertheless, meerkat groups did split up more frequently when signal transmission seemed reduced, indicating that these adjustments have limitations. In our study, animals only changed their vocal behaviour in response to indirect environmental effects influencing individual spacing. This suggests that there might be a cost to calling at higher rates and that remaining at closer distances to conspecifics might be a more efficient way to cope with changes in signal perception when conditions allow it. However, because the difference in vegetation density in our study might not have been so high as to greatly affect sound attenuation, and due to the correlational nature of our study, it is possible that these findings have a different cause. For example, increased vegetation density will also affect predator detection, which in turn can affect group cohesion or the fission-fusion dynamics of a species [48]. To fully evaluate the effects of different environmental conditions, the amount of sound attenuation during each condition should be measured. While in mobile species, manipulating the overall environment is impossible, sound playbacks or manipulating food availability might allow to investigate the effects of the different direct and indirect environmental effects with greater precision.

As has been suggested for indirect effects of noise, which can lead to an increase in individual stress and influence an animal's physiology [10], indirect effects through increased spacing might have a much stronger effect on the population than direct effects. If these conditions

persist and inter-individual distances are constrained for long periods of time, selection pressure on signalling will increase. This could lead to an increase in signalling frequency as shown here, or to structural changes of a signal (including signal amplitude, duration or modulation). Whether changes in signal frequency or signal structure are effective to counter constrained inter-individual spacing, likely depends on the structure of the original signal. Increasing the call rate for monosyllabic, short signals, such as a meerkats' close call, might be more effective than increasing signalling frequency for combinatorial signals or signals composed of multiple syllables, where additional repeats might lead to an alteration of the perceived meaning [52,53]. In our study we focused on changes in call rate and did not include changes in the amplitude or the structure of a signal. However, individuals might use a combination of different adjustments to improve signal transmission and different combinations might be more efficient depending on the original signal and the current environment. For example, olive baboons have been shown to adjust both the duration of calls as well as the frequency of signalling depending on habitat openness [6]. Finally, mammalian signalling has long been regarded as rather fixed with little vocal flexibility [54,55]. Our results confirm that of previous studies [2,9], and suggest that mammals can actually adjust their signalling behaviour to dynamic changes in their environment. Knowledge about these adjustments and potential trade-offs will allow to better understand how specific signals evolve and are maintained and as a consequence, how and whether animals relying on acoustic signalling will be able to cope with changes in the environment, be they natural or human induced.

## Supporting information

**S1 File. Additional information on results of the models fitted for the different analysis.**
(DOCX)

**S2 File. Additional analysis on change in group speed with regards to the different environmental conditions.**
(DOCX)

## Acknowledgments

We thank the Kalahari Research Trust for the permission to work at the KRC and the neighbouring farmers to work on their farms. Furthermore, we are grateful to Dave Gaynor, Tim Vink for the organization of the field site and their input on the field work, and the KMP managers Chris Duncan, Laura Meldrum, Lyndsey Marris, Sky Bischoff-Mattson and all the present and former KMP volunteers for maintaining the habituation and basic long-term data collection on the meerkats. We also acknowledge Inês Gonçalves, Denise Camenisch, Coline Muller and Megan Wyman for their assistance in collecting the vocal recordings and Richard Young for providing the NDVI data. Finally, we thank Sabrina Engesser, Elisabeth Gall and Bart Kranstauber, for comments on earlier versions of the manuscript.

## Author Contributions

**Conceptualization:** Pauline Toni, Gabriella E. C. Gall, Marta B. Manser.

**Formal analysis:** Gabriella E. C. Gall.

**Funding acquisition:** Tim H. Clutton-Brock, Marta B. Manser.

**Methodology:** Pauline Toni, Gabriella E. C. Gall.

**Project administration:** Tim H. Clutton-Brock, Marta B. Manser.

**Resources:** Tim H. Clutton-Brock, Marta B. Manser.

**Supervision:** Marta B. Manser.

**Writing – original draft:** Pauline Toni, Gabriella E. C. Gall.

**Writing – review & editing:** Pauline Toni, Gabriella E. C. Gall, Tim H. Clutton-Brock, Marta B. Manser.

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
