## [Decision Letter · Decision Letter 0]

6 May 2020

PONE-D-20-07291

Signalling adjustments to direct and indirect environmental effects on signal perception in meerkats

PLOS ONE

Dear Dr Gall,

Thank you for submitting your manuscript to PLOS ONE. After careful consideration, we feel that it has merit but does not fully meet PLOS ONE’s publication criteria as it currently stands. Therefore, we invite you to submit a revised version of the manuscript that addresses the points raised during the review process.

Both reviewers found your study interesting and worth publishing pending revisions. Please consider all their  comments and suggestions and provide detailed responses. I encourage you to follow  the suggestion to broaden the scope of the discussion, and to flag the limitations related to the correlational aspect of the results.

We would appreciate receiving your revised manuscript by Jun 20 2020 11:59PM. To enhance the reproducibility of your results, we recommend that if applicable you deposit your laboratory protocols in protocols.io, where a protocol can be assigned its own identifier (DOI) such that it can be cited independently in the future. For instructions see: http://journals.plos.org/plosone/s/submission-guidelines#loc-laboratory-protocols

We look forward to receiving your revised manuscript.

Kind regards,

David Reby, PhD

Academic Editor

PLOS ONE

Journal Requirements:

2. In your Methods section, please provide additional location information of the study area, including geographic coordinates for the data set if available.

Reviewers' comments:

Reviewer's Responses to Questions

**Comments to the Author**

1. Is the manuscript technically sound, and do the data support the conclusions?

Reviewer #1: Partly

Reviewer #2: Yes

2. Has the statistical analysis been performed appropriately and rigorously? 

Reviewer #1: Yes

Reviewer #2: Yes

3. Have the authors made all data underlying the findings in their manuscript fully available?

Reviewer #1: Yes

Reviewer #2: Yes

4. Is the manuscript presented in an intelligible fashion and written in standard English?

Reviewer #1: Yes

Reviewer #2: Yes

5. Review Comments to the Author

Reviewer #1: The Manuscript entitled “Signalling adjustments to direct and indirect environmental effects on signal perception in meerkats” represents an interesting attempt to describe how environmental changes may shape sociality and vocal behavior in a wild population of meerkats.

A lot of work has been done on this wild population, but this MS provides new insights about meerkats’ adaptability to their unpredictable natural environment.

The Manuscript is overall well written and structured, but an improvement is necessary before publication. For this reasons I suggested the Editor to consider this MS for publication, after a revision.

Introduction

All the first part of the Introduction section is centered around the difference between “direct” and “indirect” impacts on signal transmission. Nevertheless the difference between the two is somewhat misleading, as presented differently in consecutive sentences (LL. 22-29).

I suggest to rewrite the first paragraph (from line 22 to 37), by focusing on the main idea that is expressed in lines 19-20. I strongly recommend to strengthen the idea that environmental characteristics may directly impact the nature of the acoustic signal itself (in both acoustic and temporal features), or indirectly impact the signaler’s vocal production (e.g. inter-individual spacing, cohesion calls, etc.).

I suggest to insert a paragraph focusing on the role of contact and cohesion calls in primates.

L.57 I think the Authors should better explain here the differences between dry and drought season.

LL. 59-61 According to Wyman et al. 2017 [26] pups begging calls may mask the adult contact calls, so that meerkats reduce their calling rate to avoid information redundancy. For the aim of this Manuscript, I think it would be important to tell the reader if pups (and how many) were present in all the sampled groups and in all the periods. As suggested by Mausbach et al. 2017, if I am not wrong, the season between October and February should be characterized by the presence of pups (<3 months old), and I would like to understand if not only the presence/absence of pups but also the number of pups/group may influence the calling rate. Have you tested for this factor too?

I also suggest to cite here the results from Mausbach et al. 2017, according to the relationship between call rate and environmental windy condition.

I think that Authors’ hypotheses are consistent with the literature and well organized according to the logic of the Introduction section.

Methods

The Method section is well organized and detailed. Here my detailed comments:

L. 87 A number in brackets should be provided for the Clutton-Brock et al. 1998 citation.

I would suggest to the Authors to switch between Table 1 and Table S1 (Supplement1). In my opinion, the readers would be more interested in visualizing the details of the data composition for this study. Environmental details for the study period may be more appropriate in the Supplemental Materials.

LL. 121-123 The authors should be more clear here. How did they estimate the nearest neighbor identity and distances through sound recordings?

LL133-141 The Authors well explained what a group splitting is (mixed-sex sub-groups being further than 100 m apart for at least 15 minutes). It is not clear, nevertheless, how the number of group split is reported in Table S1. In Table S1 the number of groups in the Group Split column is equal to the Total number of groups (except for Dry season 2013/2014, that is dropped of one unit: this means that one group did not split in 2013/2014?). If I am not wrong, the Authors tested the frequency of group splits in the different seasons, as the % of the observation days in which the splits were observed. I think that the number of days of split over the total number of days of observation should be provided in the table or in the text.

When there were group splits did you implement the number of groups tested in the models, considering the both the group and the sub-group as separate units, or still considered the group of origin as a single group? The fact that animals are more likely to split in the drought season, consequently being further than 100 m apart for longer periods, should emerge also from the analysis of the time spent in each distance category. This is not apparently emerging from the results. Did you consider split groups as different groups in the Dirichlet regression?

Results

I suggest to pay attention to the way in which the Authors called the Supplementary Materials. Table S1 is present in both Supplement 1 and Supplement 2, thus being confounding for the reader.

I found Figure 1 very helpful for the reader. I suggest to modifying the Figure, by insert two more graphs: one for the relationship between the call rate and the presence/absence of pups (and eventually the correlation between the call rate and the number of pups) and the Figure S1.

Models provided results for the differences between Wet and Dry conditions and Wet and Drought conditions. I suggest the Authors to test the differences between Dry and Drought conditions in the different models, for example by using the multcomp package in R, to test whether significative differences emerge also for the factors in the two different conditions.

Discussion

I think that the Authors well discussed the results at the light of their hypothesis and that they should focus more on the direct (on call rate) and indirect (on group cohesion) effects of the environment on acoustic signals.

However, the discussion lacks a comment on the relationship between call rate and the presence of pups, also at the light of the findings of previous works on the same population, explaining why meerkats may cope with environmental as social factors affecting vocal emission.

LL 263-267 I think that the differences between dry and drought conditions affecting the frequency of group splits are not well discussed here.

The Authors stated that individuals are less able to maintain cohesion when signal perception is directly affected by the amount of individual spacing between group members, thus expecting a gradual increase in call rate and in inter-individual distance from the wet to the dry and drought seasons. The results are only partially in line with this scenario. How do the Authors explain the different trend observed in the dry season? By providing the results of the comparison between dry and drought season, as suggested before, the Author could also provide more explanation about the observed variation. The hypothesis that fission-fusion dynamics may explain the Authors observations is, in fact, interesting, but it is not completely clear how to read the emerging differences in the group split in relation to the dry or drought condition. Are the Authors stating (LL274-275) that in the dry condition the groups are more dispersed that in the wet condition, while in the drought season they split, and form smaller sub-groups in which the individuals disperse over longer distances? Why this dispersion did not emerge from the distance analysis?

Reviewer #2: Review of MS: Signalling adjustments to direct and indirect environmental effects on signal perception in meerkats

I have reviewed the paper by Toni et al.

I thought this was a well written and interesting MS that investigated the extent to which meerkats can flexibly modify their group coordination behaviour when faced with both direct and indirect environmental effects on signal perception.

I found the analyses to be solid and the results convincing. My comments are therefore mainly aimed at the discussion of the results which I feel could be broadened slightly.

Specifically, I found the discussion to be very meerkat-centric and hence would encourage the authors to zoom out slightly in the discussion and contextualise the results more broadly. How do their results compare and contrast with similar findings in other species and what are the implications of this? I don’t envisage an extensive re-writing here, just a paragraph or two would be useful in understanding how the meerkat results fit in to what is generally known about vocal mitigation of environmental effects on signal perception in animals.

In line with this, it might also be useful to comment on what light these findings may be able to shed on the ongoing discussion surrounding mammalian vocal flexibility. I appreciate this is not the focus of the paper, but nevertheless such results provide further support against the notion that mammal vocal production is hardwired and confirms that relatively dynamic changes in the social or ecological environment are responded to with equally dynamic changes in vocal production.

Lastly, perhaps some discussion regarding the correlational nature of the findings and the potential drawbacks of this could be important. Furthermore, if the authors have considered following up their observational results with experiments it might be worthwhile outlining what form such experiments might take.

Minor points:

1) Why did the authors not also include “group” as a random factor when investigating what factors influence group split propensity?

2) L186: Can you elaborate exactly what you compared in the LRT tests when assessing the significance of each of the fixed effects in the final model.

3) L203: Can you provide some quantification of variability around the mean for the differences in call rate when pups were present and absent?

4) L34-36: This is an oddly phrased sentence, perhaps consider revising

5) L35-37: You use “thus” twice in quick succession.

6) L198: Change to “Increasing dryness substantially affected…”

6. PLOS authors have the option to publish the peer review history of their article (what does this mean?). If published, this will include your full peer review and any attached files.

Reviewer #1: No

Reviewer #2: No

---

## [Author Response · Author response to Decision Letter 0]

19 Jun 2020

Signalling adjustments to direct and indirect environmental effects on signal perception in meerkats PONE-D-20-07291

Response to reviewers 

Reviewer #1

The Manuscript entitled “Signalling adjustments to direct and indirect environmental effects on signal perception in meerkats” represents an interesting attempt to describe how environmental changes may shape sociality and vocal behavior in a wild population of meerkats.

A lot of work has been done on this wild population, but this MS provides new insights about meerkats’ adaptability to their unpredictable natural environment.

The Manuscript is overall well written and structured, but an improvement is necessary before publication. For this reasons I suggested the Editor to consider this MS for publication, after a revision.

Introduction

All the first part of the Introduction section is centered around the difference between “direct” and “indirect” impacts on signal transmission. Nevertheless the difference between the two is somewhat misleading, as presented differently in consecutive sentences (LL. 22-29).

I suggest to rewrite the first paragraph (from line 22 to 37), by focusing on the main idea that is expressed in lines 19-20. I strongly recommend to strengthen the idea that environmental characteristics may directly impact the nature of the acoustic signal itself (in both acoustic and temporal features), or indirectly impact the signaler’s vocal production (e.g. inter-individual spacing, cohesion calls, etc.).

Response: As stated in the first paragraph (L17-42), direct environmental effects on signal transmission refer to effects that will affect the signal itself through sound attenuation or masking, while indirect effects refer to situations where the signal transmission is affected by imposing constraints on other behaviours on the signaller or receiver that will prevent optimal transmission (L20-L22). For instance, a reduction of food availability will constrain foraging behaviour and result in an increase in inter-individual spacing. The important difference as mentioned is that in the case of direct effects individuals can move freely, while in the indirect case, animals are constrained in their movement (L29-L30).

I suggest to insert a paragraph focusing on the role of contact and cohesion calls in primates.

Response: We have added more information on the function of contact calls in mammal, including primate and bird species, though we do not think a full paragraph is necessary. L43-L50: “Contact calls play a major role in group signalling in many mammal and bird species (19,20), and are either used to maintain group cohesion (21,22), or to space out group members to avoid competition (23). Consequently, animals occupying environments with unpredictable rainfall, such as savannas or half deserts, and forming highly social and cohesive groups, e.g. baboons (Papio), vervet monkeys (Chorocebus pygerythrus), meerkats (Suricata suricatta), dwarf mongooses (Helogale parvula) or pied babblers (Turdoides bicolor) to name a few, should have adapted their contact signalling to a variety of environmental conditions, whereby the specific adaptations may differ depending of the specific function of the contact calls in a system.” 

L.57 I think the Authors should better explain here the differences between dry and drought season.

Response: Here we define drought to be a temperature and precipitation extreme that lasts untypically long. We added additional information in the sentence to clarify our definition (L66-L67).

LL. 59-61 According to Wyman et al. 2017 [26] pups begging calls may mask the adult contact calls, so that meerkats reduce their calling rate to avoid information redundancy. For the aim of this Manuscript, I think it would be important to tell the reader if pups (and how many) were present in all the sampled groups and in all the periods. As suggested by Mausbach et al. 2017, if I am not wrong, the season between October and February should be characterized by the presence of pups (<3 months old), and I would like to understand if not only the presence/absence of pups but also the number of pups/group may influence the calling rate. Have you tested for this factor too?

Response: Wyman et al. 2017 found that the presence of pups rather than the absolute number of pups or the ratio of pups to adults was important to explain the effect of pups on the close call rate of adult meerkats. In addition, the sample sizes for the different pup numbers across the different seasons were quite low, which can lead to problems in the analysis. Therefore, we decided to control only for the presence of pups rather than the number of pups. Testing the effect of pups on call rate was not the purpose of this study. 

We reformulated sentences stating we included pup presence in our analyses to make it clearer that this is not what we aimed at testing (L82-L84). In addition, we do now include more detailed information on the number of pups per group per period (Table 2)

I also suggest to cite here the results from Mausbach et al. 2017, according to the relationship between call rate and environmental windy condition.

Response: We included a citation of the Mausbach et al 2017 paper in L25 and L53.

I think that Authors’ hypotheses are consistent with the literature and well organized according to the logic of the Introduction section.

Response: Thank you

Methods

The Method section is well organized and detailed. Here my detailed comments:

L. 87 A number in brackets should be provided for the Clutton-Brock et al. 1998 citation.

Response: Done (now L100)

I would suggest to the Authors to switch between Table 1 and Table S1 (Supplement1). In my opinion, the readers would be more interested in visualizing the details of the data composition for this study. Environmental details for the study period may be more appropriate in the Supplemental Materials.

Response: We agree that providing details on the data the analyses are performed on would be informative, and now include Table S1 as Table 2 in the main text. However, providing environmental information, seems also necessary to us to justify our categorization between wet, dry and drought conditions, and was stressed by a reviewer from a previous submission. We therefore left Table 1 in the main text. 

LL. 121-123 The authors should be more clear here. How did they estimate the nearest neighbor identity and distances through sound recordings?

Response: Distances to the nearest neighbour were estimated visually by the observer who followed the meerkat to record its vocalizations, and whenever a change in nearest neighbour distance category (‘0-2 m’, ‘2-5 m’, ‘5-10 m’ and ‘>10 m’) occurred, this was noted. Due to the relatively small distances, the distance categories provide fairly accurate data. We added this information in L139-L142.

LL133-141 The Authors well explained what a group splitting is (mixed-sex sub-groups being further than 100 m apart for at least 15 minutes). It is not clear, nevertheless, how the number of group split is reported in Table S1. In Table S1 the number of groups in the Group Split column is equal to the Total number of groups (except for Dry season 2013/2014, that is dropped of one unit: this means that one group did not split in 2013/2014?). If I am not wrong, the Authors tested the frequency of group splits in the different seasons, as the % of the observation days in which the splits were observed. I think that the number of days of split over the total number of days of observation should be provided in the table or in the text.

Response:

Table 2 (formerly Table S1) provides the number of groups and individuals used for each period and for each analysis performed for each period. The numbers of groups analysed change between analysis as not all data was available for each group.

For the group split analysis the response variable was the presence or absence of a group split during a visit to a group (L201-205) and not the percentage of splits for each group. We controlled for variation between groups by adding group ID in the random term. L202: “For this we used a GLMM with a binomial link function, with the presence or absence of a split on a given observation day as the response variable, environmental condition as well as total group size as explanatory variables and group identity as random term.” In the results section, we provide values of predicted average % visits out of the total number of visits, extracted from our GLMM. The information on the number of splits per number of group visits is given in the results L262-266.

When there were group splits did you implement the number of groups tested in the models, considering the both the group and the sub-group as separate units, or still considered the group of origin as a single group? The fact that animals are more likely to split in the drought season, consequently being further than 100 m apart for longer periods, should emerge also from the analysis of the time spent in each distance category. This is not apparently emerging from the results. Did you consider split groups as different groups in the Dirichlet regression?

Response: Indeed, due to the way the data is collected, group splits will not be visible in the data from the focal follows (vocal data and nearest neighbour distance data). When a group splits during an observation session (which even during the drought conditions was a rare event), the observer will stay with the part of the group pertaining to the dominant female. In addition, as a group split is only a group split when more than one individual is missing (otherwise it is more likely a single lost individual or a dispersal event), all individuals in the observed group/subgroup will have nearest neighbours within the usual distance range. Note that we only had information on the nearest neighbour and not overall distance between all group members. To clarify we added this information in the section title for the data collection (L 136-141). 

Results

I suggest to pay attention to the way in which the Authors called the Supplementary Materials. Table S1 is present in both Supplement 1 and Supplement 2, thus being confounding for the reader.

Response: Thank you. We fixed it and renamed the supplementary tables with a continuous numbering, and adjusted it in the results section of the manuscript.

I found Figure 1 very helpful for the reader. I suggest to modifying the Figure, by insert two more graphs: one for the relationship between the call rate and the presence/absence of pups (and eventually the correlation between the call rate and the number of pups) and the Figure S1.

Response: As mentioned above, our objective was to investigate the effect of different environmental conditions on contact call rate and not to investigate the effect of the presence of pups on contact call rate (this has been done in great detail in Wyman et al 2017). Furthermore, our sample size is to small to attempt to analyse the effect of pups on contact call rate with regards to different environmental conditions. Accordingly, we did not include a figure of pup presence on contact call rate. 

Similarly, while we did test the effect of environmental conditions on the condition of meerkats, this was done to show that especially the drought condition affected meerkats negatively. Nonetheless, it is not the main focus of the study and rather an additional analysis. Accordingly, we only included the most important results in Figure 1. 

Models provided results for the differences between Wet and Dry conditions and Wet and Drought conditions. I suggest the Authors to test the differences between Dry and Drought conditions in the different models, for example by using the multcomp package in R, to test whether significative differences emerge also for the factors in the two different conditions.

Response: We now provide the comparison between the different conditions in the results section and Supplement 1. 

Discussion

I think that the Authors well discussed the results at the light of their hypothesis and that they should focus more on the direct (on call rate) and indirect (on group cohesion) effects of the environment on acoustic signals. However, the discussion lacks a comment on the relationship between call rate and the presence of pups, also at the light of the findings of previous works on the same population, explaining why meerkats may cope with environmental as social factors affecting vocal emission.

Response: As mentioned previously, the objective of this study was to investigate the effect of different environmental conditions on contact call rate. We did not aim to investigate the effect of the presence of pups on contact call rate. We find similar results as Wyman et al 2017 on the effect of pups on overall call rate. We agree that it would have been interesting to discuss the effect of pups, if we had been able to test the interaction between pup presence and environmental condition. However due to our sample size we could not attempt to analyse this and we would not expect a difference.

LL 263-267 I think that the differences between dry and drought conditions affecting the frequency of group splits are not well discussed here.

Response: We added a sentence to clarify the distinction, as indeed the comparison between dry and drought conditions while being discussed was not directly mentioned (L326-L329).

The Authors stated that individuals are less able to maintain cohesion when signal perception is directly affected by the amount of individual spacing between group members, thus expecting a gradual increase in call rate and in inter-individual distance from the wet to the dry and drought seasons. The results are only partially in line with this scenario. How do the Authors explain the different trend observed in the dry season? By providing the results of the comparison between dry and drought season, as suggested before, the Author could also provide more explanation about the observed variation. The hypothesis that fission-fusion dynamics may explain the Authors observations is, in fact, interesting, but it is not completely clear how to read the emerging differences in the group split in relation to the dry or drought condition. Are the Authors stating (LL274-275) that in the dry condition the groups are more dispersed that in the wet condition, while in the drought season they split, and form smaller sub-groups in which the individuals disperse over longer distances? Why this dispersion did not emerge from the distance analysis?

Response:

No, we expected an increase in call rate from dry to wet to drought seasons; and an increase in distance to the nearest neighbour in the drought compared to both the wet and dry seasons (L78-L80). The results are in line with our prediction. We reformulated our prediction to make it clearer. We now provide the comparison between the dry and the drought condition in the supplement. 

No, we describe how under the drought (and not the dry) conditions, meerkats were observed foraging within similar distances to their nearest neighbour as in other (wet and dry) conditions, but that small subgroups were foraging further and further apart, eventually resulting in a group split. With the analysis on distance to the nearest neighbour, we investigated the distance at which the focal individual was foraging from its immediate neighbour during that observed segment, and that did not appear to change. We do not have data on the positions of all members of the group, hence our analyses did not allow the investigation of all group members dispersion. We made some changes to make our point clearer.

Reviewer #2

Review of MS: Signalling adjustments to direct and indirect environmental effects on signal perception in meerkats

I have reviewed the paper by Toni et al. I thought this was a well written and interesting MS that investigated the extent to which meerkats can flexibly modify their group coordination behaviour when faced with both direct and indirect environmental effects on signal perception.

I found the analyses to be solid and the results convincing. My comments are therefore mainly aimed at the discussion of the results which I feel could be broadened slightly.

Specifically, I found the discussion to be very meerkat-centric and hence would encourage the authors to zoom out slightly in the discussion and contextualise the results more broadly. How do their results compare and contrast with similar findings in other species and what are the implications of this? I don’t envisage an extensive re-writing here, just a paragraph or two would be useful in understanding how the meerkat results fit in to what is generally known about vocal mitigation of environmental effects on signal perception in animals.

Response: We broadened our discussion and further compare and contrast our finding with previous studies in other species (L340-350).

In line with this, it might also be useful to comment on what light these findings may be able to shed on the ongoing discussion surrounding mammalian vocal flexibility. I appreciate this is not the focus of the paper, but nevertheless such results provide further support against the notion that mammal vocal production is hardwired and confirms that relatively dynamic changes in the social or ecological environment are responded to with equally dynamic changes in vocal production.

Response: We added a sentence to this effect at the end of the discussion L381-383.

Lastly, perhaps some discussion regarding the correlational nature of the findings and the potential drawbacks of this could be important. Furthermore, if the authors have considered following up their observational results with experiments it might be worthwhile outlining what form such experiments might take.

Response: We now discuss the drawbacks of the correlational nature of the findings and suggest possible future experiments for more precise findings (L357-359). 

Minor points:

1) Why did the authors not also include “group” as a random factor when investigating what factors influence group split propensity?

Response: As mentioned L200-203 group identity was included as a random factor

2) L186: Can you elaborate exactly what you compared in the LRT tests when assessing the significance of each of the fixed effects in the final model.

Response: Done (L215-L216)

3) L203: Can you provide some quantification of variability around the mean for the differences in call rate when pups were present and absent?

Response: Done (L235-L238)

4) L34-36: This is an oddly phrased sentence, perhaps consider revising

Response: We reformulated the sentence (L37-L39)

5) L35-37: You use “thus” twice in quick succession.

Response: We removed a “thus” (L39-L40)

6) L198: Change to “Increasing dryness substantially affected…”

Response: Done (L229)

---

## [Decision Letter · Decision Letter 1]

14 Aug 2020

Signalling adjustments to direct and indirect environmental effects on signal perception in meerkats

PONE-D-20-07291R1

Dear Dr. Gall,

We’re pleased to inform you that your manuscript has been judged scientifically suitable for publication and will be formally accepted for publication once it meets all outstanding technical requirements.

Kind regards,

David Reby, PhD

Academic Editor

PLOS ONE

Additional Editor Comments (optional):

Reviewers' comments:

Reviewer's Responses to Questions

**Comments to the Author**

1. If the authors have adequately addressed your comments raised in a previous round of review and you feel that this manuscript is now acceptable for publication, you may indicate that here to bypass the “Comments to the Author” section, enter your conflict of interest statement in the “Confidential to Editor” section, and submit your "Accept" recommendation.

Reviewer #1: All comments have been addressed

Reviewer #2: All comments have been addressed

2. Is the manuscript technically sound, and do the data support the conclusions?

Reviewer #1: Yes

Reviewer #2: Yes

3. Has the statistical analysis been performed appropriately and rigorously? 

Reviewer #1: Yes

Reviewer #2: Yes

4. Have the authors made all data underlying the findings in their manuscript fully available?

Reviewer #1: No

Reviewer #2: Yes

5. Is the manuscript presented in an intelligible fashion and written in standard English?

Reviewer #1: Yes

Reviewer #2: Yes

6. Review Comments to the Author

Reviewer #1: The Authors have adequately addressed my previous comments. I thank the Authors for the text integrations and for replying to my comments with clear and precise explanations, that completely fulfill my requirements. I feel that this manuscript is now acceptable for publication.

Reviewer #2: I have re-reviewed the MS by Toni et al.

My previous issues and concerns have been adequately addressed and I am therefore happy to recommend publication.

Upon second reading I found some additional changes that could be made to aid clarification and readability.

L24: Change “with” to “including”

L31: Change “he” to “they”

L217: Change “did” to “implemented” or something similar.

320 – commas not needed.

L345: Reference here?

Plus change to: “sympatric-living”

L349: I would put this clause at the end of the sentence, rather than the middle to aid readability.

L355-357: you have “indicating” twice in quick succession. Consider using a different word.

L362: rather than highlighting about different “causes” maybe use the term “causal mechanism”?

L363: “dynamics”

L384-385: I found this sentence a little difficult to follow. It needs to be explicitly stated what your findings confirm.

7. PLOS authors have the option to publish the peer review history of their article (what does this mean?). If published, this will include your full peer review and any attached files.

Reviewer #1: No

Reviewer #2: No

---

## [Editor Report · Acceptance letter]

18 Aug 2020

PONE-D-20-07291R1 

Signalling adjustments to direct and indirect environmental effects on signal perception in meerkats 

Dear Dr. Gall:

I'm pleased to inform you that your manuscript has been deemed suitable for publication in PLOS ONE. Congratulations! Your manuscript is now with our production department. 

Kind regards, 

on behalf of

Dr. David Reby 

Academic Editor

PLOS ONE